# Assembling a global health image: Ethical and pragmatic tensions through the lenses of photographers

**Arsenii Alenichev**[1]*, **Sonya de Laat**[2], **Nassisse Solomon**[3], **Halina Suwalowska**[1], **Koen Peeters Grietens**[4], **Michael Parker**[1‡], **Patricia Kingori**[1‡]

**1** Ethox Centre, Wellcome Centre for Ethics & the Humanities, University of Oxford, Oxford, United Kingdom, **2** Faculty of Health Sciences, McMaster University, Hamilton, Canada, **3** The Schulich School of Medicine & Dentistry, Western University, London, Canada, **4** The Socio-Ecological Health Research Unit, Institute of Tropical Medicine Antwerp, Antwerp, Belgium

☯ These authors contributed equally to this work.
‡ MP and PK also contributed equally to this work.
* arsenii.alenichev@ndph.ox.ac.uk

**Data Availability Statement:** According to the research protocol for this study approved by

## Abstract

### Background

Recently, global health has been confronting its visual culture, historically modulated by colonialism, racism and abusive representation. There have been international calls to promote ethicality of visual practices. However, despite this focus on the history and the institutional use of global health images, little is known about how in practice contemporary images are created in communities, and how consent to be in photographs is obtained.

### Methods

We conducted semi-structured interviews with 29 global health photographers about the ethical and practical challenges they experience in creating global health images, and thematically analysed the findings.

### Findings

The following themes were identified: (1) global health photography is undergoing a marketing transformation and images are being increasingly moderated; (2) photographers routinely negotiate stereotypical and abusive tropes purposefully sought by organisations; (3) local scenes are modified, enhanced and staged to achieve a desired marketing effect; (4) 'empowerment' is becoming an increasingly prominent dehumanising visual trope; (5) consent to be photographed can be jeopardised by power imbalances, illiteracy, fears and trust; (6) organisations sometimes problematically recycle images.

### Interpretation/Discussion

This research has identified practical and ethical issues experienced by global health photographers, suggesting that the production cycle of global health images can be easily

Oxford University Central University Research Ethics Committee (CUREC), the qualitative data set should not be made public in order to maximize confidentiality of the research subjects. As such, research subjects agreed to participate in this study specifically because of my (AA- the primary author who conducted all interviews) ethical commitment of maximising confidentiality of their responses, which was also reflected in signed consent forms obtained from each individual participant. Making the data set publicly available may result in the public sensitive data disclosure, loss of employment and damaged professional relationships, alongside other risks for the subjects. If you have questions, please contact the above-mentioned ethics board at ethics@medsci. ox.ac.uk referencing the approval number 'R84443/RE001: Ethical issues in Global Health visuals'.

**Funding:** This work was supported by Wellcome Trust (grant 221719 to AA, PK, MP and HS). The funders had no role in study design, data collection and analysis, decision to publish, or preparation of the manuscript.

**Competing interests:** The authors have declared that no competing interests exist.

abused. The detected themes raise questions of responsibility and accountability, and require further transdisciplinary discussion, especially if promoting ethical photojournalism is the goal for 21st century global health.

## Introduction

*Photographing is essentially an act of non-intervention. . .in situations where the photographer has the choice between a photograph and a life, choose the photograph. The person who intervenes cannot record; the person who is recording cannot intervene* [1]

Susan Sontag (1977)

Sontag's argument about the role of the photographer and the ethics of (non)intervention is well documented and long-debated among photojournalists and academics [2, 3]. However, little attention has been paid to the position and ethical experiences of photographers working in global health–a field that stems from colonial medicine and its visual culture, has historically reinforced visual prejudices and stereotypes [4], and currently relies on both photojournalistic [3, 5] and marketing practices [6, 7] for visual communication. While photojournalism may tend to view itself––even if this is contested––as calling for non-interference and objective and reflective depiction of local realities, a marketing approach aims to create competitive and emotion-triggering advertisements with the goal of engaging the audience and producing behavioural effects.

Continuing the legacy of exploring global health issues through experiences of various frontline workers, in this article we zoom into the lived experiences of global health photographers and the ethical and pragmatic challenges they confront. We reflect on how global health photographers, by virtue of their role, directly confront the pain and suffering of others and are uniquely positioned to represent and document individuals and communities on behalf of global health institutions. In such contexts, the argument that photographers maintain a detached and distanced relationship with the world does not always hold in practice, as global health photographers actively interfere with local communities and navigate complex moral landscapes. By zooming into the lived experiences of such photographers, we showcase a spectrum of issues pertaining to the social production of informed consent and global health images that can be easily overlooked. Recognising that photography is a powerful tool that can result in good or in creating or perpetuating harms if used improperly, in this article we primarily explore the latter, offering case studies for global health theory and practice. In doing so, the paper 'demystifies' the creation of global health visuals, expanding the discussion concerning visual ethics and their role in the (de)colonisation of global health [8–11].

## Materials and methods

In this article, we employ terms such as **Global North and Global South** to map photographers and interventions, while acknowledging the limitations that comes with such heuristic categorisations. The primary reason for using this dichotomy was to signify the prior colonial relations and the neo-colonial integration of the South into the world markets. We define **global health** as a broad umbrella term for various health-related interventions conducted primarily in the Global South, which are led by Northern institutions or significantly dependent on Global North funding. By **global health visuals** we mean visual depictions of people,

material items, landscapes, and communities used to illustrate interventions, campaigns, collaborations, diseases and aspects of treatment. **Global health photographers** are professionals of various backgrounds who produce such images. By **tropes** we mean the visual scenarios that are reproduced in seemingly unrelated images, which includes examples of **'poverty porn'**, the visual trope that emphasises individual suffering and nudity for a visceral effect, and **'white saviours'**, images of international aid workers in communities that emphasise white people as care providers, usually in the centre of an image, surrounded by Black and Brown people, usually children.

## Identification and recruitment of informants

This study was approved by the ethics board at the University of Oxford (CUREC: R84443/ RE001). We identified potential informants (1) through our professional global health networks, (2) by searching for 'global health photographer' in a search engine that retrieved the professional websites of the photographers, and (3) by contacting the communication teams of global health organisations and asking for a photographer referral. Our inclusion criteria were that respondents should be aged 18 or above and have a willingness to talk about their professional experiences with global health photography in a semi-structured interview. A convenience and snowball sampling approach was employed, and more than 100 photographers were contacted using the ethics-board approved recruitment email draft and a participant information sheet about this study. If potential informants expressed interest in participating, video calls were scheduled in MS Teams prioritising the availability of the informants. During recruitment a strong emphasis was placed on the voluntary nature of participation and maximisation of anonymity of the responses. The primary researcher (AA) conducted all recruitment, consent and data collection for this study.

## The consent process

At the start of every call, participants were provided with a verbal version of the participant information sheet and had an opportunity to ask questions about this study. Once informants were happy with the answers, written informed consent of the approved format, tailored for online interviewing, was obtained from every participant. As part of this consent process, the primary researcher (AA) shared the screen with the consent file blank and read the consent statements aloud and clarified statements when needed. Participants were asked to explicitly say whether they agreed or disagreed with such statements (10 in total, e.g. 'I had a chance to ask questions about this study and received satisfactory answers'). Consent forms included two optional statements asking participants if the call could be recorded and if the quotes could be used anonymously in the outputs. All informants agreed to be quoted and recorded. Participants were informed that they could withdraw their data after the interview, until 1 September 2023. After each interview AA sent a signed and scanned copy of the consent form to every participant.

## The data collection process

An external recording device was used to capture the conversation and no video was recorded. The topic guide was iteratively modified and updated after each interview, and AA discussed emerging themes and questions with the other members of the team as part of the regular debriefings. The final version of the topic guide covered the following themes and sub-themes inquiring into participants':

• professional background and how they ended up doing global health photography;

- thoughts on the history and current state of global health photography;

- experiences with global health organisations (clients) and ways in which participants create images in accordance with the work assignments;

- experiences of working in communities and obtaining consent from the photography subjects, with a particular focus on ethical and pragmatic problems; and

- wishes and advices for global health organisations creating imagery and other global health photographers.

Following the conclusion of interviews, participants were offered £20 compensation vouchers for their time.

## Data analysis

Audio recordings were transcribed by services endorsed by the University of Oxford, and the audio recordings were then deleted. Transcripts were then edited for clarity and broad categories to identifiable data and participants were applied, (e.g. 'a photographer from the Global South', 'Global Health organisation', 'West Africa') to maximize anonymity when presenting the findings, and imported into Nvivo (v 1.7.1) for iterative coding. Iterative thematic analysis was used to analyse the findings, and emerging themes were constantly identified among the interviews conducted by AA and discussed with the team of authors consisting of specialists of various backgrounds including global health history, visual anthropology and global health communication. In accordance with the ethics approval, only AA had access to the data set, he provided the team with anonymised case-studies and interview summaries, and received guidance on subsequent data analysis steps. Maximising anonymity and validity, informants were re-contacted to seek their explicit approval for the use of sensitive quotes. Data was stored on encrypted servers and will be destroyed at a later date in accordance with the standard operating procedures at the University of Oxford.

## Results

### Participant characteristics

In total, 29 informants were interviewed between March and July 2023. The interviews averaged 60 minutes in length and all were held in English. Informants had diverse professional backgrounds, among them: professional and self-taught photographers, journalists, storytellers, and marketing and communications specialists, with all of them employed by Global Health organisations to produce images. All informants worked in the Global South, produced images that were published online by global health organisations, and were experienced in obtaining informed consent for photography from people and communities in the Global South. Of the 29 photographers interviewed, 12 were white, 12 were from the Global South and 11 were women, with women from the Global South being the most underrepresented group. This sample reflects the reality of Global Health photography, which historically has been dominated by people from the Global North, mostly men, on 'parachute' assignments of taking photos and flying back [12]. This is reflected in the following quote from a Black female photographer from the Global North:

And also, the thing I want to add to my first statement is from the opposite side of the lens, you know, the people who were receiving me were also surprised to see someone like me on this side of the camera. They're so not used to that and they would be like, 'Who are you?', What are you doing?'. They want to know everything about me and how I got there and it

usually resulted in a very open conversation and relationship because they immediately also feel a little safer if it's somebody who they think looks a little bit more familiar than a complete stranger who doesn't speak their language, doesn't look like them, doesn't understand the culture.

Informants listed several overarching reasons for starting doing global health photography: (1) telling the truth about local contexts and problems and depicting them as accurately as possible, (2) humanising and giving voice to people and communities in difficult circumstances (3) assisting the global health initiatives in their cause, and, as well as (4) a passion for photography and (5) the traveller lifestyle and an opportunity to learn from other people and contexts.

The following themes emerged from the analysis of the interview transcripts:

## The rise of marketing global health photography and highly moderated imagery

According to the informants, Global Health photography originally followed photojournalistic practices in the 1980s—often problematic ones—which is reflected by tropes of 'suffering subjects' and 'poverty porn' that emphasised the suffering and misery of people, which in the 1990s was followed by the 'white saviour' trope glorifying the ways in which health care workers and volunteers provided humanitarian help to local communities, vividly resonating with earlier colonial imagery. With the rise of media and activism all such tropes become actively criticised, which resulted in the emergence of the global health marketing and communication teams responsible for the creation of ethically sensitive imagery. However, numerous informants, particularly those with photojournalistic backgrounds or those following photojournalistic principles, were concerned that global health photography, which they saw as historically a photojournalistic practice, is now undergoing a marketing transformation, resulting in photos that are 'sanitised' and 'misrepresenting reality':

Now organisations start saying, 'Well, we cannot show this, we cannot show that.' And then now they are, they start giving you more strict instructions. Then that's the point I told them: Okay, sorry, you have so many restrictions **you cannot show the reality, you know.** You guys have to go to the remote area and see what the reality is. (Photographer: Global North, emphasis added)

Informants attested that the marketing approach to Global Health photography is now a force to be reckoned with, and it has two main functions: to (1) avoid abusive photojournalistic practices that might negatively affect the organisations, and to (2) create attention-grabbing images that can compete with mainstream advertising, such as banner ads, and generate revenue for the organisation.

Because agencies put down a certain style of photography that they would like to follow. . . pictures must be bright, there should be no shadows, you know, no black, no light and shade. . . it has to be clean, uncluttered backgrounds. . . evenly lit. **That artistic kind [of photo] and parameters are not something that I could fulfil without staging things.** And, to be honest with you, I mean, to be very, brutally honest, a lot of the documentary photography for nonprofits is actually advertising work. Because they need the picture for advocacy, for the websites to get more funding, to influence people. And it's, **it can be problematic.** (Photographer: South Asia, emphasis added)

In another quote, a photographer explains that the number of restrictions in briefs directly interferes with photographers' artistic vision and photojournalistic principles:

> And other photographers told me it's, very limiting creativity, right, because, many people they're trained to be storytellers, they're trained to be quote-unquote 'proper photographers,' but in the end just with the sheer number of instructions that they're given it's like, it's almost, you're producing stock images—and often you're producing stock images—just because you, you pretty much give away your rights to images so it's, it's quite sad, I'd say. (Photographer: South East Asia)

## Briefs and negotiating power in global health imagery

Informants reported that the marketing and communication teams often create 'briefs' containing instructions on what images are needed for a given assignment. Photographers reported that they try to routinely negotiate such prescriptive assignments, as exemplified by the following quote from an established local photographer:

> So, for me, a client, even if they provide a brief, **it's my duty to also educate them**. . . And there was one time where [an international NGO] wanted photographs of [malnourished] kids, you know they had to remove the kid's clothes and put her on this old-fashioned scale, and I'm like: **I'm not taking that picture.** So yeah, those are some of the challenges. . . The client says 'I'm paying you, I want this [picture from you]'. . . (Photographer: Southern African region, emphasis added)

Several informants attested that it is problematic for them that global health communication departments are often located in the Global North, and that in creating the briefs they are often 'disconnected' from the local realities of the South.

> I think a lot of people who are in [global health communications departments] they really like, don't have any understanding of actually making the pictures or being a photographer, unless you're a photographer first, it's kind of difficult, but it really seems like there's a massive disconnect there sometimes. It seems like they don't understand what they're asking for. I'm not sure how that happens, but it does. (Photographer: Global North, emphasis added)

In the following quote, a West African photographer provides another example of briefs as vectors of stereotypes as they were tasked to provide an impossible photo of a baobab tree in a Sub-Saharan context:

> [Communication teams] can give you any brief, you know, sometimes they are sitting somewhere out there, they have no idea what's on the ground. But, you know, the truth is that if you send them a compelling story, they will use it. Maybe if you are beginning, you want to follow the brief to the letter because you don't want to make a mistake. But I think that at a certain level they actually come to you, or they should come to you because of your competence. . . For example, I had to photograph some baobab trees and the client sends a photo from Madagascar, you know, and that type of baobab is different. **I Instantly I told the client, 'This photograph, you either have to use AI or I have to go to Madagascar. Across sub-Saharan Africa, you will never see that. '** So, then the client knows and then you show them the kind of pictures that are more realistic. It's, communication, isn't it? (Photographer: West Africa, emphasis added)

However, challenging the briefs was not always enough to pursue a locally sensitive photography. In a quote below, an early-career female photographer from West Africa experienced a situation where a global health organisation sent someone else to take images after she refused to:

*Interviewee*: We had a campaign here. So, they were looking for images of children who are suffering from malnutrition. You know, the usual image of how Africa has been portrayed and for me it was also personal, because it's not just people looking like they are suffering in that way. For instance, there has been a drought, but you cannot tell if people are experiencing it: [local people] might skip meals, like maybe they might eat once a day, but you may not necessarily see that kind of [visible] suffering. I'm not gonna just go and take those images, because they requested me to. **I really have to challenge those requests and I have done so, yeah. And there are instances where after I did** *laughs***, they came themselves to take those images.**

*Interviewer*: If I understood correctly, you challenged the brief. So they sent someone else to take the images that they initially wanted?

*Interviewee*: Yeah. That's what happened. I mean, it almost created this divide between their team and my team, for quite a while. But like, I think they came to understand that sometimes it's important [to take local concerns seriously]. You know, but I think they're also grappling with the idea of raising funds.

In the quote below, a female photographer from South Asia showcases how briefs fuel stereotypes, and could also result in direct physical harm to both photographers and photography subjects:

You know, this one NGO wanted me to do an [assignment on Leishmaniosis], which is this parasitic disease and it's very prevalent in [communities] and an NGO wanted me to take photos and they were like, 'Oh, if you see the fly, make sure you take a photo of the fly.' And I was just like, 'So you want me to get bitten and also get Leishmaniosis, because that's how you get it? Okay. So, and I'm just like, '**Wow, like, you actually want me to go near a fly and not run in the opposite direction if I see one?**' It's just, like, beyond my understanding, you know, so this was, like, the most absurd one. But they, they also say, 'Oh, you're working in a [refugee community] but you have to show women's faces.' That is not possible. If a woman shows me her face, the photograph her, she gets killed. It's that serious, they kill their women. If their picture comes out, they actually kill their women in the communities. It's happened in the past and it's just so sensitive that one has to be exceptionally careful because you don't want to put your subjects in danger, no matter what. **Like, that's number one, you have to protect them at any cost, you know, and that's what the consent is for. But then it's, like, double standards, like, 'No, make sure to get the consent but also get their face pictures,'** I'm not okay with, like, so you're just like, 'Okay, you know, it's not easy,' yeah.

In the following quote, a photographer elaborates on their experiences with a malnourishment assignment and actual local realities that a global health organisation refused to document, resulting in a moral breakdown experienced by the photographer:

I was asked by a [global health organisation] to go somewhere in [East Africa] to a refugee camp, to photograph a malnourished kid. And I arrived and I found out that in that

particular time, there was no malnourishments. And there were only five babies in the maternity wards which were malnourished at that time, but that was, kind of, normal, you know, that would happen in any given situation. And I called [the communications team] and said, like: 'Listen up. There is no malnourishment here. **So how are we, how on Earth are we going to campaign on malnourishment when it's not existent?**' Then their reply was, 'Okay, go back, go to the **ward and find the best looking baby and make a story on that baby.** But, you know, make sure blah blah blah.' I didn't do that. And, near the refugee camp, there were a lot of Somali refugees which were not able to enter, the, the refugee camp because there were with so many. And, and I, said to the organisation, 'Now listen up, you know, there's all these Somalis who are not, who cannot enter and, you know, that's a story, you know. These people need help because they have got nothing.' **And then they said, their reply was, 'Yeah, but we cannot campaign on Somalians because we know Somalians are not very likeable, you know.**' And that was, that was the strongest thing ever happened to me in my career. Where, where, where I thought like, 'This is so fucked up. This is so fucked up.' Yeah. What, am, am I, you know, am I one of them? But I was so gutted by the whole situation, I felt so bad, and that I decided not to work for, for, for years, for NGOs. There was a break in my career. (Photographer: Global North, emphasis added)

## Staging, re-enacting and enhancing scenes

Informants reported that they are aware that they influence local scenes just by being present in communities, and there are at least three ways in which photographers and local teams could actively interfere with the scenes in order to complete photo assignments:

(1) **Staging,** when a scene is carefully assembled for the purpose of taking a photo. Below, for instance, is a quote exemplifying ways in which scenes are purposefully assembled and orchestrated:

You [photographer] go on a day you know things will be happening. For maternal health, vaccinations or reproductive health, for example, [in an East African Context] most rural hospitals have a healthy baby club which meets on a particular day of the week. Teams always plan hospital trips for those days, because then it provides the illusion (or, emphasises the fact) that the care being provided comes courtesy of the organisation. That way consent is often implicitly linked to provision of care. You give things. Soap, wash tubs, sometimes salt. Field teams usually travel with a few such items in the boot of their white land cruisers, and there's a giveaway after the media collection is done. (Photographer: Global North).

In another quote, a photographer explains how, in their experience, the images of vaccine administration and smiling kids are staged:

I got a really big project documenting children getting vaccinated and it was so complicated. **They said, 'Okay, you have to photograph children who are getting vaccinated but they can't be crying in the pictures.' That was number one. Number two was, 'You can take some pictures with the needle and some not with the needle.' So then what are you really photographing, kids getting vaccinated or not getting vaccinated, you know what I mean?** I would be, like, 'Don't inject the kid yet. Let me take the pictures first because once you do that the kid's gonna cry for the next 10 minutes and there will be no

pictures,' it's like pulling your hair. . . My [family member] also was a consultant with the [major international organisation], not a photographer, but he helped me understand. He said, 'Don't consider it photojournalism. Consider it corporate photography, because that's what they want. They want pictures like poster child and, like, laughing and smiling and that's not the reality.' And, you know, it's actually like I'm always in this moral dilemma because these children are in such poor, poor situations and I am expected to make them smile and laugh and it's just so fake because these children aren't wearing shoes and you're telling them, 'Laugh,' Laugh at what? They're generally not in a good mood. If they naturally smile and they laugh because they're kids and they're playing, that's fine. **But in a lot of cases [global health organisation staff members] make them laugh and they make them smile. Sometimes I have to talk to the kids and be like, 'Oh yeah, what's up?' But, you know, it's all staged. It's all corporate photography.** (Photographer: South Asia, emphasis added)

Or, for instance, another example of staging involved nurses being asked to pretend to be patients:

If you're going **in for a global health company, you have to stage it, but it's not like you're not lying.** You're not like going in there saying, 'This doesn't happen.' You're saying, 'Hey, can your nurse be a patient?' It happens in the United States all the time. It's the same thing, it is advertising. (Photographer: Global North, emphasis added)

(2) **Re-enactment,** when a routine action is repeated specifically for the photographer, as exemplified by the following quote:

I mean, first and foremost I think of myself as a journalist. And I will never convey in an image something that's not accurate. The only staging I would do is, like, maybe this activity doesn't typically happen on Tuesdays and I'm there on Tuesdays. **But the activity typically happens, you know?** (Photographer: Global North, emphasis added)

(3) **Enhancement,** when a scene is modified for a desired positive or negative effect.

If it's a commission, if it's non-journalistic, **I may occasionally intervene in small ways, like, 'there's trash in the background, can someone pick it up? Or, 'Can we, can we get this bucket out the way?'** I will generally not direct the scene or construct the scene, and say, 'You be here, you be there,' and all that. I tend to work with what's in front of my lens and, and the light that I have. The exception would be that sometimes in the NGO space, you need images of a particular type, like, you need eye contact and obviously direction is necessary. And you would say, '**Okay, alright now look at me,' or, 'Turn a little bit to the left and look at me while you do whatever it is.**' But I think that's a different kind of image and it's also an image where it's clear that there is engagement between the person in the photo and the photographer. It's obvious, right? And I see images that are purporting that there is no engagement, **or images that are purporting to be documentary, and they're actually not. And I have a problem with the latter, actually**. (Photographer: Global North and South, emphasis added).

In the quote below, a photographer provides another example of a scene enhancement along the lines of a 'tender mother' trope, also often referenced as the 'pieta' (or Madonna and child) cliché:

Imagine the scene, like, the cholera beds that are very narrow and you see a kid that is lying on the bed, and the mother that is doing the interview, so the [media crew member] will ask her, 'Oh, can you, can you take the kid in your arms and balance him a bit or, or can you do again this moment where you remove the hair in front of his eyes, you know. **So, there is faking in the sense of capturing a tender image of a mother with a kid, but not staging in the sense you have cholera.** (Photographer: Global North, emphasis added).

In the following quote, a photographer shares their experience of enhancing the scenes by giving people N95 masks:

I was on an assignment to [produce a cover for a global health document] I was documenting [infectious disease] and, you know, took photos of [people in a rural hospital in Central Asia], and then when I brought back the images to [global health organisation], the [organisation's] communication teams told me 'Well I'm sorry, we can't use that image." I said, 'Why?" 'Oh, because [the people in the images] are not wearing N95 masks.' 'Wait a minute,' you know, that clinic they didn't have N95; you guys think every clinic in the world in remote places have N95 mask, but they don't. So what they have is just surgical masks and they wear surgical masks. And they insisted: 'No, no, we cannot show that.' **From that experience, I had to bring N95 mask with me, okay [and give it to people for photos]. But, it's a kind of setting up: it's not a fact. It's not reality.** [For global health organisations], that was a dilemma, you know: reality versus ideal, you know, as in what they want to do. (Photographer: Global North, emphasis added).

One of the key listed reasons for such active interference is pragmatic: according to the informants, in many cases the global health organisations send photographers to the communities for a very short time—sometimes just several hours—while, according to the informants, for a more genuine reflection several days or even weeks are needed to capture 'everyday lived realities', resulting in a situation where photographers are forced to actively interfere with local scenes in order to deliver assignments:

Sometimes it's because the time is so short, like they [clients] might give you a day or two to capture those images. So, you are forced to kind of stage those images, instead of just stay there and just follow the natural process. Because time is so limited, you have to have to dictate the people how to act. And I, I don't think that's a way to go. (Photographer: East Africa)

## A new trope: Images of empowerment

Reflecting on the global rise of the marketing approach in global health photography and influencing of scenes, informants reflected that 'images of empowerment', which have been part of international development photography for decades, is now becoming a fully-fledged trope that is being actively enacted in the communities, focusing specifically on showcasing images of smiling and healthy people in challenging environments as signifiers of a global health intervention. Numerous respondents regarded this to be as dehumanising as older tropes such as 'suffering subjects' and 'white saviours':

If it's not real, if it's not the reality, like, I don't, I'm not going to do that [photography]. But, in terms of empowerment: what does it really mean to be empowered? I mean, now it's got this weird colonial thing. It's, like, to be empowered, you have to be given power. Who gave

you the power? Who gave you the information? And it's the white saviour who give it to you. So, you know, "Oh look. I'm filled with the holy spirit,' you know. (Photographer: Global North)

I mean, they don't need your empowerment. They just need some money actually. (Laughing) They have so much power, that's what I mean about being able to survive with their living, they don't need your empowerment, and they are the strongest people we could ever possibly know. (Photographer: Global North)

[The empowerment trope] is still in the same line [as] white saviourism in which, it's kind of like, how can I say? It's like we're overly protecting a person, saying, 'Oh, look, how resilient he is.' There is some sort of looking down on this person in a way because you're like, 'Oh, look, you know, he made out a living, out of selling peanuts'. (Photographer: Global North)

In the following quote, the photographer showcases the concrete practical issues stemming from the conflict between imaginations in briefs to demonstrate empowerment and satisfy a narrative an organisation is promoting, and the local reality:

I was working with one of the organisations, and the communication person was with me, and she was saying, 'Look, these are the guidelines we've been given for the kind of photographs of empowered women, what they should look like,' you know, 'And can you please try and follow that?' I said, 'Look, I can try and follow it. I mean, I don't have anything against it. But I can follow it only as far as portraiture goes, you know, slightly lower angle, look, make the person look a bit empowered. And I have no problem with that. But I can't do it when I'm photographing them doing something, you know.' For example, I was photographing a community and their livelihoods are coming from needlework. **And I said, I can't photograph them doing needlework with them looking up into the sky, you know, it's impossible. They're going to prick their fingers, you know, and it's impractical.** (Photographer: South Asia, emphasis added)

According to the informants, images of empowerment also require peculiar technical and practical preparations, as evident from the quote below:

Going back to empowerment, the, yeah, the trope of empowered people I think as I say like as a way of kind of expressing the industry's need to continue to produce visuals even though it has been told that the visuals it produces are immoral, that results in the empowered subject. But it's, it's entirely an illusion, I mean I can, I can show you how to set up an image of empowerment: So you use something that has become much more available and cheap, it is a neutral density filter. It darkens the image without affecting the balance of tones with it. Once you darkened the scene you then use a flash to light the scene up, thus brightening it up again but now because you've got a, a dark image you can use as a very shallow depth of field, you can open your aperture very, very high, because you don't have much light coming into the camera. So, you can have like a single image of a single person in the middle of the image, and then you've got all that space around you which you can use for text. (Photographer: Global North)

Informants also acknowledged that there is the possibility of a split between actual meaningful empowerment and empowerment as a staged visual practice, also stemming from the abovementioned problem of the limited time photographers can work in the communities:

And you can also look for, not just the aesthetics of composition and so on, but you can also look for the small moments that are what I like to, kind of, call, and sort of, shared bridges of humanity. And so, I think that those things are all part of empowerment, but when you have the time and the space to represent people in a dignified way. And then there's empowerment in the sense of, 'We need a portrait of somebody and make them look empowered,' you know, which may or may not be the actual reality that those people are in. And I think that's more when one gets into that, sort of, staged scenario that we've come in, I don't understand what the contexts, you know, I haven't even had a chance to say 'hello' to the leaders of this village and explain what I'm doing here. **And, you know, we've just come in, 'Here's somebody that is a beneficiary,' and, you know, 'We need an empowering looking portrait of her, you have five minutes, take a picture'**. (Photographer: Global North and South)

In the quote below, a female photographer from West Africa reflects on global health empowerment and its visual culture:

The word is so overused, like, everywhere you go you just hear empowerment, empowerment. It becomes really tiring. I remember a friend of mine also she wrote some, sort of, essay for a programme, and she sent to me and I was reading it and she said, 'I want to empower women.' And I'm like, 'Who are you? You know, who are you to empower women? Were these women not doing these things for centuries before you arrived? So, what are you going to do for them, right now?' So, I think the word empowerment is even condescending sometimes. Like, I put this trick that will make you a magician (Laughs). It's a little weird for me, I avoid that word **because I feel, like, maybe you shouldn't say empowerment you probably should find a word where you want to say that you are adding to what is already there**. (Photographer: West Africa)

Moreover, one photographer expressed an opinion that 'Sometimes, out of the fear of poverty porn or being accused of poverty porn, photographers actually refuse to tell the story that is there, the true story', highlighting the challenge of finding ethical way of showcasing local challenges and suffering.

## Informed consent, inequalities, and trust

All informants experienced issues and challenges with regard to obtaining consent for photography.

As such, consent forms were typically written in the main official language of the country, and the translation and interpretation constituted an issue for photographers, as evident from the following quote:

There's also been the challenge of communicating what information is on consent forms. So, all these organisations have their consent forms. Most of these forms are in English and there are moments or times that local people are not able to read and understand the information on these forms. So, I have to go through, explain to them what exactly the form is saying and if they agree to that, then I go ahead. So sometimes it can be quite hectic because you have very limited time to make these images. Consent I think is the most important thing for me, if I don't someone's consent and I'm photographing the person, I feel like I'm stealing from that person. (Photographer: West Africa)

Moreover, according to informants, written consent forms became increasingly complex, involving dense legal language and, leading to various issues, including the perception that signing a form for photography is a mandatory part for receiving care:

Bringing out of a legal document, when many people are non-literate first of all, makes it even more, just of a farce. We might have it in a local, like, the dominant language. But maybe we're in a region that speaks [different language] you know what I mean? It's a page long. And it's written, even if the language is translated, the type of, just the verbiage is [too dense]. **I actually think it does more harm to people to produce a legal document, ask them to sign it than it does not to do that. Because it's intimidating.** The big issue with [legally dense] consent, you've got people who are used to queuing up for services from whatever the provider is that we are accompanying. And many countries, in many of the cases, you know, people are used to having to provide some sort of national ID to receive services and whatever it is. And it's just part of the, it's part of the process. You go to the front of the queue. You show your ID. You pay, you sign something. You get your ART or whatever it is. **And if I ask them to sign something, it implies that being photographed is part of what they need to do or receive care.** And that's my belief, personally. (Photographer: Global North, emphasis added)

The following exemplifies a similar concern of how chronic social inequalities fundamentally affect the consent process for photography:

[Here, in some communities] the literacy rate less than 50% and that's anyone who can read or write their name are also considered literate. And a lot of people can't even write their name, so how are you going to explain to them all the aspects that are present in the consent form? And then a lot of times, you know those are communities who, if you get them to try and sign something, they expect something in return. They think that they're going to get money, or they're going to get funds, they're going to get some kind of help and that's why their signature was required. And sometimes I would photograph one person and I would get their consent and the other person would get upset that you get, got them to sign something now they're going to get something in reward. (Photographer: South Asia)

Photographers are sometimes perceived as doctors or someone who is better able to help because they inquire about the lives of people and their challenges, as evident from the following quote:

The project [on chronic infectious disease patients] was so challenging: the village where they are living was really difficult to reach. They were really depressed and had no access to health care. **They always believed that I was a doctor, even though I told them I'm a photographer. But I accepted that because this is how they treat me, this is how they trust me. Otherwise, I would lose that access**. . . And the second thing about health: other infections and malaria are also there. Hopefully I'm not, I'm not infected. (Photographer: South East Asia, emphasis added)

Moreover, according to photographers, some express distrust and fears pertaining to giving away a signature due to past traumatic experiences, as evident from the following quote:

Especially with marginalised communities and groups of people and poor people who have not had the good fortune of having good education and lawyers and stuff like that, any

piece of paper terrifies. And signing on a piece of paper terrifies people. 'What is this? Am I signing away my little piece of land to somebody?' Because that's what has happened before, you know, the money lenders have given money and made them sign on blank forms and taken away the land. (Photographer: South Asia)

Informants also highlighted that a one-off consent is not enough to prove the ethicality of the photography use, as evident from the case study below:

I followed someone living with HIV for about six years, documenting [their] life each year and how it changed. And this was pre-like, the internet explosion. And, so I document all of the stuff. I handed it over to my client for campaigns. It travelled a bit. **It was ethically done, I had all the consent forms, everything. You know, they had signed it all, like they'd agreed. I had a relationship over six years, so they knew very well.** It wasn't like I was a stranger coming in. Then it got to the point where it became very easy for people in low-income communities to start 'googling' themselves. And this person gets an opportunity. And she looks herself up. Or maybe a friend looked her up. . . and all those HIV, AIDS pictures come up. There's a very good chance that most of the people on her community didn't know her status. **So, she came into a hospital hysterical, kind of, saying, 'What the hell's going on?' And, I mean, typically, we had every right to use those images in that context.** And, but, in this position, like, it very clearly was no longer the most beneficial thing for this patient, I had to go through this process of then contacting all of these multiple organisations and asking them to pull down all these pages on their website. Thank goodness we managed to do it. (Photographer: South African Region, emphasis added)

## Recycled images and the alienation of local contexts

Several participants reported that images they produced for international global health organisations were subsequently recycled for other purposes, and saw this as potentially ethically problematic. Below is an example of how an image from West Africa travelled to depict global health in East Africa:

Sometimes you work for really big organisations, they have offices in all these countries, all over the world. So, I think one time [my] image I'm talking about was used in [Office in East Africa]. **So officially for African ones, it's the same black skin. Like, they kind of use the images transferably.** So, images for a campaign or for a cause in [West Africa] are used for something else––in another African country, because people can't really tell.

The same photographer attested that the repurposing of images into photobank items can cause significant problems locally:

I happened to photograph one girl [patient with a serious disease], I think she was three years old. So she was with the mother, we sought consent, you know. And I photographed her, but then she passed on, I think after a week, but. . . **so now, this girl is no longer, no longer alive and for me as a photographer, I really would have loved if her images could have been left out for whatever campaign or whatever purpose the images are going to be used for.** But unfortunately, I still keep seeing her image being used for [campaigning] by the client and it's quite difficult. Because these images end up in should I say, media banks of these organisations and reversing the process is quite, it, it's not an easy task sometimes. (Photographer: West Africa, emphasis added)

## Discussion: Demystifying the social production of global health imagery

In recent years, global health organisations have been confronting prevailing abusive and dehumanising photography, especially the tropes of 'suffering subjects' and 'white saviours', and attendant visual prejudices, that have their roots in the formal colonialism and the ongoing neocolonial integration of the South into the world markets [13–17]. In order to deal with these images in an ethical manner, current recommendations suggest that organisations have to showcase consent obtained from people to be in photos, and promote respectful depictions and representations of people and their communities [8]. This paper, empirically rooted in the lived experiences of global health photographers, suggests that while the current recommendations offer excellent ethical benchmarks for moderating the ethical use of imagery by organisations, they remain insufficient to ensure the ethical production of global health imagery.

With focus on the **use** of images and providing **evidence of consent**––practices that unfold at the level of organisations––it is easy to overlook the layers of ethical complexities pertaining to ways in which images are socially created in real-world contexts, and how consent is obtained. Based on the experiences of photographers, we argue that the marketing approach to global health photography increasingly involves staging, re-enacting, and enhancing stereotypical scenes, to which photographers often react with resistance. Such interferences often result from photographers having very limited time and space to complete the assignments. In such restricting settings, briefs become powerful carriers of stereotypes and normativity that participate in the re-production, amendment and proliferation of stereotypes. As such, positive imagery such as 'empowered subjects', which is currently promoted as a heuristic solution to prior abusive tropes [18], can be effectively staged or enacted for the purpose of taking a positive representational photo, risks becoming, nearly literally, an elite global health capture [11] informed by mainstream marketing principles. This indicates that 'empowered subjects' could be emerging as yet another potentially dehumanising visual trope to depict people and their life-worlds that stems from the cultural logic of white saviours as enables of power and providers of care. The default portrayal of people in challenging environments along the overly positive lines of empowerment vividly resonates with the tokenistic 'washing' common in mainstream advertising and corporate social responsibility programs [19–21], that are now being increasingly applied to depict interventions in the Global South [22, 23], raising questions about the features of contemporary global health visual culture and its possible future directions.

Moreover, according to photographers, photography subjects might in some–possibly many– situations be giving their consent to be in photos in ways that are influenced by known factors such as unwarranted trust [24], structural coercion [25, 26] and other manifestations of inequities and disparities common in global health. As a result, photography subjects are not necessarily provided with sufficient information with regard to what they are agreeing to, nor how their images might travel, part of which could be photographers' overlooking of ethical dimensions. The practice of recycling imagery for use in disease, intervention or geographical contexts other than that in which it was originally captured requires careful transdisciplinary examination by the broader global health community, especially with regard to accountability and responsibility.

Moving forward, a push for ethical global health photojournalism and meaningful photojournalistic collaboration, as promoted by the MSF [27] and various global health initiatives, could be a more ethical way to approach 21st century global health photography. It likely comes with economic sacrifices: in the age of social media, global health visuals are competing with mainstream advertising for attention from viewers to yield emotional responses and donations, which partially explains why a more marketing approach to global health emerged in the first place. Perhaps, the utopian solution to ensuring the absolute ethicality of global

health photography would be to promote the development of national health systems in a resilient, sustainable adaptive way, so that the images of suffering or overcoming it will not be needed at all to attract funding and ensuring the delivery of care.

## Conclusions: The global health visual culture at the crossroads

In this study, we replied to the growing international call to critically reflect on the visual culture of global heath, by zooming into the personal and professional experiences of global health photographers. We showcased how the creation of global health visuals is a morally and ethically complex process unfolding in ordinary contexts, whereby photographers routinely deal with representational dilemmas and challenges shaped by the prescriptive briefs detailing what images a photo assignment should yield. This, however, structurally creates openings for the (de)humanisation of photography subjects and a jeopardising of the validity of obtained informed consent. Reflecting on such case studies, we argued that the major source of the ongoing tension in global health photography stems from the broader conflict between photojournalism as an attempt to truthfully and reflectively depict local realities, and marketing photography aimed at creating attention-grabbing sensationalist imagery, creating opportunities for coercion and staging of scenes for global health visuals. We argued that the staging of scenes along the lines of 'empowerment' signifies the emergence of the visual trope in global health, a successor of older suffering subject and white saviour tropes, requiring a careful transdisciplinary examination. Taken together, such case studies have to be translated into the practice if our goal is to decolonise global health and build more equal postcolonial societies wherein careful and sensitive depictions of people is a basic human right.

## Limitations and self-reflexivity

There are several factors that might have influenced the data collection and analysis. First, the study might have attracted respondents who already had critical experiences that could be shared anonymously, particularly by global health photographers of photojournalistic and storytelling backgrounds. As such, experiences of marketing global health photographers and perspectives of global health marketing and communication teams might differ. In order to yield a more complex picture future ethnographic studies could be built around shadowing photographers and observing the work in the communities, and documenting the experiences of photographed people. Second, given that the interviews were conducted in English, interviews were biased towards native and non-native English speakers with good internet connection, excluding a significant portion of the non-English speaking global health community that remains offline. Third, this article provides a broad overview, and future studies could specifically inquire into more localised experiences, allowing for cross-contextual comparison. Finally, only the lead author conducted the interviews, thus personal biases might have played out in reporting the findings from this study, which is a peculiarity of any qualitative research.

## Acknowledgments

We would like to thank all interviewed photographers for sharing their stories, and two anonymous peer reviewers for their helpful feedback. We are grateful to Marla Shaivitz for suggestions, guidance and critical review of the manuscript.

## Author Contributions

**Conceptualization:** Arsenii Alenichev, Sonya de Laat, Nassisse Solomon, Halina Suwalowska, Koen Peeters Grietens, Michael Parker, Patricia Kingori.

**Data curation:** Sonya de Laat, Nassisse Solomon, Halina Suwalowska, Koen Peeters Grietens, Michael Parker, Patricia Kingori.

**Formal analysis:** Arsenii Alenichev.

**Funding acquisition:** Michael Parker.

**Investigation:** Arsenii Alenichev.

**Methodology:** Arsenii Alenichev, Patricia Kingori.

**Supervision:** Michael Parker, Patricia Kingori.

**Writing – original draft:** Arsenii Alenichev, Sonya de Laat, Nassisse Solomon, Halina Suwalowska, Koen Peeters Grietens, Michael Parker, Patricia Kingori.

**Writing – review & editing:** Arsenii Alenichev, Sonya de Laat, Nassisse Solomon, Halina Suwalowska, Koen Peeters Grietens, Michael Parker, Patricia Kingori.

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
