## [Decision Letter · Decision Letter 0]

26 Sep 2023

PGPH-D-23-01698

Of staging, suffering and empowerment: Ethical and pragmatic experiences of photographers in creating global health visuals

Dear Dr. Alenichev,

Thank you for submitting your manuscript to PLOS Global Public Health. After careful consideration, we feel that it has merit but does not fully meet PLOS Global Public Health’s publication criteria as it currently stands. Therefore, we invite you to submit a revised version of the manuscript that addresses the points raised during the review process.

Comments from the Editor: In addition to addressing Reviewer comments, please address the following:

a) Description of methodology could be more robust by ensuring adherence to the CORE-Q32 checklist for rigorous reporting of qualitative studies.  See https://academic.oup.com/intqhc/article/19/6/349/1791966 

b) "Global North" and "Global South" are imperfect terms. Please describe/define how you are using them at the beginning of the article.

We look forward to receiving your revised manuscript.

Kind regards,

Heather Haq, M.D., M.H.S.

Academic Editor

Journal Requirements:

1. Please send a completed 'Competing Interests' statement, including any COIs declared by your co-authors. If you have no competing interests to declare, please state "The authors have declared that no competing interests exist".

2. Please provide a/amend your detailed Financial Disclosure statement. This is published with the article. It must therefore be completed in full sentences and contain the exact wording you wish to be published.

a. Please clarify all sources of funding (financial or material support) for your study. List the grants (with grant number) or organizations (with url) that supported your study, including funding received from your institution. 

b. State the initials, alongside each funding source, of each author to receive each grant.

c. State what role the funders took in the study. If the funders had no role in your study, please state: “The funders had no role in study design, data collection and analysis, decision to publish, or preparation of the manuscript.”

3. Since your data is not available for proprietary reasons, please explain via email why the data is not available. Please also include the contact information for the third party organization that should be contacted should other researchers want to request access to this data and please include the full citation of where the data can be found. We also request that you verify with us via email that any researcher will be able to obtain the data set in the same manner that the you have obtained it. If you feel you are unwilling or unable to adhere to this policy, please explain your reasons by return email and your exemption request will be escalated to the editor for approval. Your exemption request will be handled independently and will not hold up the peer review process, but will need to be resolved should your manuscript be accepted for publication. One of the Editorial team will be in touch if they require more information.

Additional Editor Comments (if provided):

Reviewers' comments:

Reviewer's Responses to Questions

**Comments to the Author**

1. Does this manuscript meet PLOS Global Public Health’s publication criteria? Is the manuscript technically sound, and do the data support the conclusions? The manuscript must describe methodologically and ethically rigorous research with conclusions that are appropriately drawn based on the data presented.

Reviewer #1: Yes

Reviewer #2: Yes

2. Has the statistical analysis been performed appropriately and rigorously?

Reviewer #1: N/A

Reviewer #2: N/A

3. Have the authors made all data underlying the findings in their manuscript fully available (please refer to the Data Availability Statement at the start of the manuscript PDF file)?

Reviewer #1: Yes

Reviewer #2: No

4. Is the manuscript presented in an intelligible fashion and written in standard English?

Reviewer #1: Yes

Reviewer #2: Yes

5. Review Comments to the Author

Reviewer #1: This manuscript insightfully introduces the readers to the complex experiences of photographers, and their subjects, in current-day global health photography. It highlights themes including the effect of marketing on photography, the insufficiency of how image consent is currently approached, and the ironic consequences of the well-intended push for “empowerment” tropes. It was an intriguing and eye-opening read. I especially appreciated the selection and depth of participant quotes that effectively convey the message to the reader. It calls for re-examining the way global health images are obtained, used, and propagated. I think it makes an important addition to the global health literature. I noted a few revisions below.

Most Important:

1. Referencing the Standards for Reporting Qualitative Research, there are a few missing methodology details. It would be important to explicitly include the following:

a. The authors’ research paradigm,

b. A reflexivity statement: while it is controversial whether one or multiple coders are sufficient in qualitative medicine, it is helpful to learn the coders’ characteristics and experiences as to help the reader understand the lens with which data was interpreted, and

c. Any limitations of the study.

2. Page 9, the last paragraph in discussion section: The proposed solution of restructuring health systems, while related, seems to be a far reach from the conclusions the authors arrived to in this research. The assertions in the last paragraph, in general, need to be toned down, or at least thoroughly explained and supported by evidence from the literature. Alternatively, they would make for a great opinion piece.

Other Revisions:

1. Introduction: Second paragraph: 1st sentence: “arguably the most precarious contexts in the world” is a very big statement. I recommend rewording or at least citing where it has been argued.

3. Some terminology that readers of various backgrounds may not be familiar with and could benefit from some explanation: “poverty porn,” “trope,” “comms,” which was used in a respondent’s quote, and “visual washing.”

4. The key reported problems with obtaining consent can be organized in a table rather than bullet points. It is an especially enlightening list of problems, and could be more easily referenced and shared in a table.

Finally, out of curiosity, when the informants were re-contacted, were the authors’ able to conduct any member checking/ informant feedback regarding the results?

Reviewer #2: The authors have presented on ethical issues in global health photography. They highlight important considerations that need addressing as part of the move to decolonise Global Health and prevent the perpetuation of stereotypes. This is work is timeous and relevant for the development of a suitable framework for Global Health photography.

6. PLOS authors have the option to publish the peer review history of their article (what does this mean?). If published, this will include your full peer review and any attached files.

**Do you want your identity to be public for this peer review?** For information about this choice, including consent withdrawal, please see our Privacy Policy.

Reviewer #1: No

Reviewer #2: No

---

## [Decision Letter · Decision Letter 1]

18 Dec 2023

Staging suffering, and empowerment in Global Health imagery through the lenses of photographers

PGPH-D-23-01698R1

Dear Dr. Alenichev,

We are pleased to inform you that your manuscript 'Staging suffering, and empowerment in Global Health imagery through the lenses of photographers' has been provisionally accepted for publication in PLOS Global Public Health.

Congratulations on this very important work and great manuscript!

Best regards,

Ashti Doobay-Persaud

Academic Editor

Reviewer Comments (if any, and for reference):

Reviewer's Responses to Questions

**Comments to the Author**

1. If the authors have adequately addressed your comments raised in a previous round of review and you feel that this manuscript is now acceptable for publication, you may indicate that here to bypass the “Comments to the Author” section, enter your conflict of interest statement in the “Confidential to Editor” section, and submit your "Accept" recommendation.

Reviewer #1: All comments have been addressed

Reviewer #2: All comments have been addressed

2. Does this manuscript meet PLOS Global Public Health’s publication criteria? Is the manuscript technically sound, and do the data support the conclusions? The manuscript must describe methodologically and ethically rigorous research with conclusions that are appropriately drawn based on the data presented.

Reviewer #1: Yes

Reviewer #2: Yes

3. Has the statistical analysis been performed appropriately and rigorously?

Reviewer #1: N/A

Reviewer #2: N/A

4. Have the authors made all data underlying the findings in their manuscript fully available (please refer to the Data Availability Statement at the start of the manuscript PDF file)?

Reviewer #1: No

Reviewer #2: Yes

5. Is the manuscript presented in an intelligible fashion and written in standard English?

Reviewer #1: Yes

Reviewer #2: Yes

6. Review Comments to the Author

Reviewer #1: Revisions to the methods were addressed, and now meet the standards for reporting qualitative research. I especially appreciated the explanation of new terminology, and found the new summary section insightful! I did find the concise list of “key reported problems with obtaining consent” in the last manuscript version to be especially helpful prior to the interview excerpts. Nevertheless, I propose no additional required revisions. Great manuscript!

Reviewer #2: (No Response)

7. PLOS authors have the option to publish the peer review history of their article (what does this mean?). If published, this will include your full peer review and any attached files.

**Do you want your identity to be public for this peer review?** For information about this choice, including consent withdrawal, please see our Privacy Policy.

Reviewer #1: No

Reviewer #2: No
